# The Effect of Farmland Transfer on Agricultural Green Total Factor Productivity: Evidence from Rural China

**DOI:** 10.3390/ijerph20032130

**Published:** 2023-01-24

**Authors:** Guoqun Ma, Xiaopeng Dai, Yuxi Luo

**Affiliations:** 1School of Economics and Management, Guangxi Normal University, Guilin 541004, China; 2Pearl River-Xijiang River Economic Belt Development Institute, Guangxi Normal University, Guilin 541004, China

**Keywords:** farmland transfer, agricultural green total factor productivity, nonagricultural labor transfer, agricultural technology utilization

## Abstract

Exploring the effect and mechanism of farmland transfer on agricultural green total factor productivity (AGTFP) in China is of great significance for exerting the effectiveness of China’s farmland transfer policy and promoting green agricultural development. Based on panel data from 30 provinces from 2005 to 2020, this paper applies a two-way fixed effects model to analyze the impact of farmland transfer on AGTFP, and the mechanism of farmland transfer on AGTFP is also investigated. We find that farmland transfer has a significant and sound promoting effect on AGTFP, with respect to multiple robustness checks; there is heterogeneity regarding the impact of farmland transfer on AGTFP in terms of food functions, and farmland transfer can promote regional AGTFP through nonagricultural labor transfer and agricultural technology utilization. When considering the fact that farmland transfer has increased China’s AGTFP, the Chinese government should continue to adhere to the farmland transfer policy, accelerate nonagricultural labor transfer, improve the level of agricultural technology utilization, and ultimately promote green agricultural development.

## 1. Introduction

Since the 21st century, China has achieved rapid growth in gross agricultural output value and agricultural products by relying on the massive use of chemical production factors, such as fertilizers and pesticides. However, this extensive growth mode with high input and low output has also led to excessive pollutant emissions, causing severe damage to the agricultural environment [1,2]. Thus, the Chinese government proposes to co-ordinate pollution control and ecological protection and promote green agricultural development [3]. Green agricultural development means reducing agricultural pollution and protecting the agricultural ecological environment while ensuring agricultural output [4]. Its essence is represented by the growth in agricultural green total factor productivity (AGTFP) [5].

In the process of improving the ecological environment and promoting AGTFP, land reform centered on the household contract responsibility system extensively mobilized the enthusiasm of farmers and promoted the rapid growth of agricultural production [6]. However, China’s long-term tension of “more people and less farmland” still exists. In order to improve the unit output of farmland, agricultural producers heavily use agricultural chemicals, such as pesticides and fertilizers, resulting in severe agricultural nonpoint source pollution and serious damage to the agricultural environment [7]. In order to improve the efficiency of farmland utilization and protect the agricultural production environment, the Chinese government has begun to implement a land system to encourage farmland transfer [8]. According to the “Annual Report on China’s Rural Policy and Reform Statistics (2005–2020)” compiled by the Ministry of Agriculture and Rural Affairs, the circulation area of farmland in China increased from 3.65 million hectares to 35.5 million hectares. Theoretically, farmland transfer can not only reduce the degree of land fragmentation and improve the scale operation of farmland [9], but it also has an influence on agricultural economic performance [10] and environmental performance [11,12], thus affecting AGTFP. This leads to the question: What is the impact of farmland transfer on AGTFP? What is its internal mechanism?

The existing studies on farmland transfer have mainly focused on the relationship between farmland transfer and economic and environmental variables, such as household income [13], poverty vulnerability [14], allocation efficiency of agricultural land [15], and environmental performance [16,17,18]. Li et al. (2019) [13] used fuzzy mathematical methods to estimate the welfare gap before and after farmland transfer in Guangzhou, Wuxi, and Chongqing, finding that the overall welfare of farmers after farmland transfer increased by 17.5%, 15.1%, and 23.5%, respectively. Liu et al. (2022) [14] found that various poverty alleviation models have different effects on poverty alleviation, among which farmland transfer is the best model. Fei et al. (2021) [15] found that the provinces in which farmland transfer occurs are more efficient for land use than those without farmland transfer. Li et al. (2021) [16] believe that the expansion of the scale of farmland caused by farmland transfer will encourage farmers to increase the use of organic fertilizer, improve soil fertility, and thus improve agricultural output and environmental performance. Wu et al. (2018) [17] found that when the farm size increased by 1%, the average fertilizer and pesticide application decreased by 0.3% and 0.5%, respectively. This is because the expansion of the scale of the farmland helps farmers to implement environmentally friendly production behavior and improve their willingness to adopt environmentally friendly technologies [19]. However, some scholars put forward the opposite view. Jiang et al. (2021) [18] point out that moderate-scale operations did not meet the policy expectation of curbing agricultural nonpoint source pollution; Mao et al. (2021) [20] also found that farmland transfer improved the overall nonpoint source pollution level of the transferred farmland.

Previous pieces of literature have mainly focused on the connection between farmland transfer and agricultural TFP (total factor productivity). Adamopoulos and Restuccia (2014) [21] found that farmland transfer can increase agricultural TFP by promoting capital-intensive technology. Zhu et al. (2011) [22] point out that farmland transfer has significantly improved agricultural TFP in areas where the farmland transfer market is highly developed. Liu et al. (2019) [23] found that farmland transfer can effectively improve agricultural technology efficiency and then improve agricultural TFP. Helfand and Taylor (2021) [11] believe that the scale expansion brought about by farmland transfer reduced agricultural TFP. Kuang et al. (2021) [10] found a significant “inverted U-shaped” relationship between farmland transfer and agricultural TFP.

There exists a lot of literature on the relationship between farmland transfer and economic growth, and environmental performance. However, on the one hand, there is still a research gap on the impact of farmland transfer on AGTFP; on the other hand, previous studies mainly analyze the impact of farmland transfer at the agricultural producers or agricultural family level, and few studies analyze the effect of farmland transfer policies from the macrolevel. Therefore, this paper uses a two-way fixed effect model to explore the impact of farmland transfer on AGTFP based on provincial panel data from 2005 to 2020. The results show that farmland transfer has significantly increased AGTFP.

The main contributions of this paper are the following: first, previous studies have mainly focused on the economic effects of farmland transfer, while limited research has addressed the environmental impact of farmland transfer. This paper considers the combined economic and environmental effects of farmland transfer and examines how farmland transfer affects AGTFP, which enriches the relevant studies. Second, most existing studies have been conducted at the agricultural producer individual level and agricultural family level, ignoring the policy implementation effect of land transfer at the macrolevel. This paper provides a trend analysis based on the empirical study of panel data from 30 provinces in China. Third, previous studies have mostly focused on the direct effect of land transfer on environmental pollution, with little attention to the important intermediary mechanism of nonagricultural labor transfer and agricultural technology utilization. This paper incorporates nonagricultural labor transfer and agricultural technology utilization into the analytical framework, reveals how farmland transfer influences GTFP, and conducts a mechanism test, which helps in finding out how farmland transfer affects green agricultural development.

The rest of the paper is assembled as follows: Section 2 introduces the theoretical analysis and research hypotheses; Section 3 presents the empirical model and methodology, including the econometric model, variable description, and data source; Section 4 reports the empirical results and analyses; Section 5 presents the discussion; Section 6 presents the research conclusions and recommendations.

## 2. Theoretical Analysis and Research Hypotheses

### 2.1. The Direct Influence of Farmland Transfer on AGTFP

Farmland transfer can directly affect AGTFP by reducing land fragmentation and expanding the operational scale. On the one hand, the fragmented farmland increases the material cost of agricultural production [24]. In the case of limited cultivated land resources, small farmers are more likely to use chemical factors to increase agricultural output, which increases agricultural pollution emissions and damages the agricultural ecological environment [25]. At the same time, small farmers have less access to agricultural production technology. They have less enthusiasm to accept green agricultural technology, which leads to the long-term use of traditional agricultural production modes with high input and low output [26,27].

On the other hand, farmland transfer will also affect AGTFP by improving agricultural production intensity. Specifically, farmers with large-scale farmland have more strength in capital and technology and have a greater willingness to adopt environmentally friendly and resource-saving agricultural technology [28,29]. Meanwhile, farmers with large-scale farmland also have advantages regarding the introduction of agricultural green production technology and the use of green production factors [30]. In addition, the scale and intensification of agricultural production are conducive to the rational allocation of land, capital, and other production factors [15], which can promote the application of agricultural green production technology, reduce pollution emissions, and promote AGTFP [31]. Accordingly, this paper puts forward research Hypothesis 1:

**Hypothesis** **1:**
*Farmland transfer can directly increase AGTFP.*


### 2.2. Indirect Transmission Mechanism of Farmland Transfer on AGTFP

In the context of rapid urbanization and industrialization, farmland transfer will encourage inefficient rural labor transfer to cities and nonagricultural industries, which will have an impact on AGTFP [32]. On the one hand, part of the income increase brought about by the nonagricultural employment of rural labor will be used for agricultural production, such as the use of agricultural machinery and the purchase of green production factors, which will promote the optimization of the agricultural production structure and then improve AGTFP [33]. Meanwhile, the nonagricultural rural labor force has more opportunities to get in touch with green production concepts and green production technology, which is conducive to promoting and applying agricultural green production technology and then improving AGTFP [34]. On the other hand, nonagricultural transfer is often characterized by high levels of labor outside of agricultural production, and a “low degree” of work continues to engage in agricultural production in rural areas [35,36]. In this case, most farmers will increase the chemical input elements to ensure the yield [37,38,39], which will reduce the quality of the farmland, increase agricultural pollutant emissions, and inhibit AGTFP.

Farmland transfer also affects AGTFP through agricultural technology. Specifically, farmland transfer will expand the farming scale of farmers and enhance their demand for agricultural technology, especially the application of agricultural machinery [40]. Meanwhile, farmland transfer can also help farmers adopt agricultural green prevention and control technologies and solve the problem of chemical pesticide residue [41,42]. The extension of agricultural machinery and agricultural green prevention technology can optimize the use of elements, which can improve AGTFP [4]. However, with large-scale mechanized farming, the input and consumption of fossil energy also increase, which will produce more air pollutants and inhibit AGTFP [20]. Thus, this paper puts forward the following hypotheses:

**Hypothesis** **2:**
*Farmland transfer influences AGTFP through nonagricultural labor transfer.*


**Hypothesis** **3:**
*Farmland transfer influences AGTFP through agricultural technology utilization.*


## 3. Empirical Model and Methodology

### 3.1. Econometric Model

Theoretical analysis shows that farmland transfer will directly affect AGTFP. In order to empirically verify the theoretical analysis, the authors propose an econometric model [10] as follows:(1)$$\;AGTFP_{it}=\alpha_{0}+\alpha_{1}FD_{it}+\beta X_{it}+\gamma_{t}+\mu_{i}+\varepsilon_{it}\;$$

Equation (1) is used to test Hypothesis 1, where $AGTFP_{it}$ represents agricultural green total factor productivity in province *i* in year *t*, $FD_{it}$ represents the scale of farmland transfer in province *i* in year *t*, $X_{it}$ indicates a series of control variables, $\gamma_{t}$ and $\mu_{i}$ denote the time-fixed effect and the regional-fixed effect, respectively, $\varepsilon_{it}$ indicates the random disturbance term, and $\alpha_{0}$, $\alpha_{1}$, and β are the parameters to be estimated.

To identify the mechanism of the role of farmland transfer in increasing AGTFP (Hypothesis 2 and 3), the authors construct the following moderating effect model, referring to the literature [43]:(2)$$\;AGTFP_{it}=\alpha_{0}+\alpha_{1}FD_{it}+\alpha_{2}FD_{it}\times Z_{it}+\beta X_{it}+\gamma_{t}+\mu_{i}+\varepsilon_{it}\;$$
where $\;Z_{it}$ denotes nonagricultural labor transfer and agricultural technology utilization in province *i* in year *t*. We use the interaction term $FD_{it}\times Z_{it}$ to identify the moderating effect of the nonagricultural labor transfer and agricultural technology utilization. If the coefficient $\alpha_{2}$ is significantly positive, it indicates the presence of a moderating effect. Further, to deeply identify the role of nonagricultural labor transfer and agricultural technology utilization, we construct the following mediating effect model, referring to the literature [34,44]:(3)$$\;Z_{it}=\theta_{0}+\theta_{1}FD_{it}+\beta X_{it}+\gamma_{t}+\mu_{i}+\varepsilon_{it}\;$$
(4)$$\;\;\;\;\;AGTFP_{it}=\delta_{0}+\delta_{1}FD_{it}+\varphi Z_{it}+\beta X_{it}+\gamma_{t}+\mu_{i}+\varepsilon_{it}$$

### 3.2. Variable Description

#### 3.2.1. Explained Variable

The explained variable in this paper is agricultural green total factor productivity (AGTFP). We use the super-efficiency SBM-DEA (slacks based measure-data envelopment analysis) model to calculate the AGTFP, for which the dynamic characteristic was measured with the GML (global Malmquist-Luenberger) index [45]. The SBM-DEA model is one of the commonly used methods to measure AGTFP [46]. It can avoid the measurement error of a subjective setting production function, can avoid the problem that the foreseen output and the undesirable output change in the same proportion, and can avoid the overestimation of technical efficiency when there is nonzero relaxation in the input or output [47,48]. The SBM-DEA model is also applicable to the measurement of a multi-input-multi-output model [49]. In this paper, the calculation of AGTFP includes eight inputs and three outputs. Therefore, the SBM-DEA model is more suitable for our work. When using the SBM-DEA model, there may be some efficient decision-making units, and it is difficult to compare these efficient decision-making units [50]. Referring to Chen and Liu (2022) [51], we use the super-efficiency SBM-DEA model to calculate AGTFP. A static efficiency value is used when considering AGTFP calculated by the super-efficiency DEA-SBM model, referring to Lv et al. (2021) [45]; we use the GML index to measure the dynamic changes in AGTFP. The GML index is used to solve the problem of there being no feasible solution for linear programming, and it can be further decomposed into agricultural green technology efficiency (AGEC) and agricultural green technology progress (AGTC) [52]. Since the AGTFP measured above is a chain index, we convert it to a fixed-base index to reflect cumulative trends in AGTFP [53]. That is, assign AGTFP in 2005 as 1, and then the actual value of AGTFP in 2006 is the product of AGTFP in the current year and AGTFP in 2005, and the actual value of AGTFP in 2007 is the product of AGTFP in the current year and AGTFP in 2005 and AGTFP in 2006, and so on.

We refer to [54,55,56] when taking the provinces as the primary decision-making units, and we selected eight factors, including land, labor, draft animals, mechanical power, irrigation, pesticides, agricultural film, and fertilizer as the input variables. The output variable is divided into foreseen output and undesirable output. The foreseen output is the total agricultural output value expressed by a constant price. The undesirable output refers to various environmental pollution, including water pollution, soil pollution, and carbon emissions. Among them, the calculation methods for water pollution and soil pollution refer to Liu and Feng (2019) [57], and the calculation methods for carbon emissions refer to Yu et al. (2022) [58]. Considering that there are many types of agricultural pollution, referring to Wei et al. (2020) [59], we combine agricultural water pollution and soil pollution into the agricultural comprehensive pollution index using an entropy weight method. The indicators and data sources for the measurement of AGTFP are shown in Table 1.

#### 3.2.2. Core Explanatory Variables

We selected the scale of farmland transfer (FD) as the core explanatory variable. The scale of farmland transfer can directly reflect the effect of the farmland transfer policy implemented by the Chinese government [15]. To make the research results more reliable, we also selected the ratio of farmland transfer to agricultural operation scale (FDA) as an alternative explanatory variable, which is measured by the scale of the farmland transfer to the total sown area of crops in the primary industry.

#### 3.2.3. Intermediary Variable

We selected nonagricultural labor transfer (FLT) and agricultural technology utilization (ATS) as the mediating variables to further examine the mechanism of farmland transfer on AGTFP. Nonagricultural labor transfer is measured by the number of employed laborers in the nonagricultural industries/the number of employed laborers in the planting industry [43]. The higher the FLT, the more pronounced the nonagricultural transfer effect of the labor force. The ATS represents the total power of agricultural machinery per unit sowing area [4]. The higher the ATS, the higher the degree of agricultural technology utilization.

#### 3.2.4. Control Variables

Referring to existing research [60,61,62], we select the following control variables: (1) Agricultural structure (INS), measured by the proportion of the added value of planting industry in the added value of the primary industry; (2) Income distribution (IND), expressed as the ratio of per capita disposable income in urban areas to per capita disposable income in rural areas of each province; (3) Rural energy consumption (EN), measured by the per capita electricity consumption of rural residents in each province and transformed by taking logarithm; (4) Financial support to agriculture (FSA), expressed as the proportion of provincial expenditure on the primary industry in total financial expenditure; (5) Dependence of agricultural products trade(OPEN), expressed as the ratio of total import and export of agricultural products to the gross value of agricultural output.

### 3.3. Data Source

We selected the panel data of 30 of China’s provinces from 2005 to 2020 (excluding Tibet, Hong Kong, Macao, and Taiwan). Among them, the input and the output data for AGTFP and the control variables were taken from the “China Statistical Yearbook” and “China Rural Statistical Yearbook” from 2006 to 2021. The data for farmland transfer scale were taken from the “Statistical Data of National Rural Economy”, “Statistical Annual Report of China’s Rural Operation and Management”, and the “Statistical Annual Report of China’s Rural Policy and Reform” from 2006 to 2021. The quantity of nonagricultural labor transfer was acquired through the “Statistical Yearbook of China’s Population and Employment” from 2006 to 2021. The data for the total import and export of agricultural products were taken from the “China Agricultural Yearbook” and the “China Agricultural Products Trade Development Report” from 2006 to 2021. Some missing data are supplemented by linear interpolation. The descriptive statistical characteristics of the variables are shown in Table 2.

## 4. Empirical Results Analysis

### 4.1. Calculation Results for AGTFP

When considering that the Chinese government issued the “Ministrative Measures for the Transfer of Rural Land Contracted Management Rights” in 2005, which expressly stipulated the principles of farmland transfer, we use 2005 as the base period to measure China’s AGTFP and the results are displayed in Table 3.

Overall, China’s AGTFP continued to increase from 2005 to 2020, with an average annual growth rate of 4.95%, mainly driven by AGTC (6.76%). It shows that the performance of China’s green agricultural development is constantly improving. From a regional perspective, the average annual growth rates for AGTFP in the western, eastern, and central regions from 2005 to 2020 are 5.74, 4.6, and 4.35%, respectively. The higher growth rate for AGTFP in the western region may be due to the relatively backward development of agricultural production. After the introduction of advanced green production technologies from the eastern and central regions, they have shown rapid growth. Table 3 shows that the AGTC in the western region is 8.88%, which is higher than that in the eastern region (5.35%) and the central region (5.79%). The central area comprises mostly grain-producing provinces. When agricultural producers carry out large-scale production, they use advanced technologies to reduce production costs. This also leads to the intensive use of input elements and a reduction in pollution emissions. Table 3 shows that the average annual growth rate of AGEC is 0.001%.

### 4.2. Regression Results for Farmland Transfer on AGTFP

We used Stata 16.0 to perform the statistical analyses and estimated the impact of farmland transfer on AGTFP. The Hausman test shows that χ^2^(6) = 16.41; the corresponding *p*-Value is 0.012, rejecting the original assumption. Thus, we use a two-way fixed effect model to estimate the impact of farmland transfer on AGTFP, and the results are shown in Table 4. The influence coefficient of FD on AGTFP is positive, and the *p*-Value of the *t*-test is less than 1%. Hence, it is reasonable to reject the null hypothesis that FD does not influence AGTFP in favor of Hypothesis 1, whereby an increase in FD improves AGTFP. This is because the scale of expansion of farmland transfer can change the production mode (with families as the production unit) and promote the intensification and scale of agricultural production. This can encourage agricultural producers to adopt green production technology, improve the utilization efficiency of chemical elements, and reduce agricultural pollution emissions. It also can reduce agricultural producers’ financing constraints, enabling them to expand agricultural production investments and reduce production costs through large-scale production, thus promoting AGTFP.

In terms of control variables, all the influence coefficients of INS, EN, and FSA on AGTFP are positive, and all the *p*-Values of the *t*-test are less than 1%. Indicating that the growth in the planting industry, the increase in rural energy consumption, and the improvement of the financial support for agriculture have all improved AGTFP. The growth in the planting industry means the scale of agricultural production improves, which can realize the optimal utilization of resources and reduce the threshold of technology adoption. The increase in rural energy consumption indicates that the use of agricultural machinery also increased, which can reduce the use of the agricultural labor force and reduce the cost. The improvement in financial support for agriculture can improve the profit expectation of agricultural producers and encourage agricultural production. Moreover, the Chinese government’s financial support for agriculture tends to promote green agriculture, which can increase the use of biological pesticides and farm manure, reduce agricultural pollution emissions, and promote AGTFP.

All the influence coefficients of IND and OPEN on AGTFP are negative, and the *p*-Values of the *t*-test are less than 1% and 5%, respectively. The increase in the urban-rural income gap will cause agricultural producers to heavily use chemical factors to increase income, which will cause agricultural nonpoint source pollution and damage the agricultural production environment. Moreover, the widening urban-rural income gap will also promote the transfer of young and middle-aged labor to nonagricultural industries, leading to the abandonment of land and a reduction in agricultural output, thereby inhibiting AGTFP. In addition, agricultural product trade will urge agricultural producers to increase the agricultural chemical factors to increase agricultural product export, resulting in increased agricultural pollution emissions and worsening AGTFP.

### 4.3. Treatment of Endogenous Problems

We take the lag term of farmland transfer as the explanatory variable and use the instrumental variable method to deal with the endogenous problem caused by the possible two-way causality between FD and AGTFP, and the interference of the random disturbance term on FD. The autocorrelation test shows that the AC value of lag 1 of FD is 0.7781, and the *p*-Value is 0.0007, rejecting the null hypothesis. It indicates that there is a positive autocorrelation between FD and the lag 1 of FD. Meanwhile, the result of the system GMM shows that the *p*-Value of the Arellano-Bond test for AR (2) is 0.278, accepting the null assumption. It shows that the second-order autocorrelation of FD is not obvious. Thus, we take the lag 1 of FD for regression, and the result is shown in Reg (1) of Table 5. It can be seen that the influence coefficient of the lag 1 of FD is positive, and the *p*-Value of the *t*-test is less than 5%, indicating that the lag 1 of FD increases AGTFP. Secondly, we use the system GMM method for regression, and the results are shown in Reg (2) of Table 5. The results show that after the lag 1 of AGTFP was included in the regression, the influence coefficient of FD is positive, and the *p*-Value of the *z*-test is less than 5%, which is consistent with the regression results in Table 4.

Thirdly, we attempt to construct an exogenous variable and use the two-stage least square method (2SLS) for regression. We refer to the practice of Chong et al. (2013) [63], which uses the average value of farmland transfer in neighboring provinces in the same year as the instrumental variable (IV). On the one hand, the neighboring regions may learn from each other when implementing the farmland transfer policy; the farmland transfer of neighboring provinces can positively affect the farmland transfer of the region [64]; on the other hand, an improvement in AGTFP in the region can expand the scale of the farmland transfer in this province but have little impact on the farmland transfer in neighboring regions [65].

Reg (3) and Reg (4) in Table 5 are the regression results using the instrumental variable. The Reg (3) shows that the influence coefficient of IV on the farmland transfer is positive, and the *p*-Value of the *t*-test is less than 1%, rejecting the null hypothesis that IV does not influence FD. It indicates that the farmland transfer of neighboring provinces has spillover effects on the farmland transfer in the region. The Reg (4)shows that the influence coefficient of FD on AGTFP is positive, and the *p*-Value of the *t*-test is less than 1%, which is consistent with the regression result in Table 4. Meanwhile, the *p*-Value of the Kleibergen–Paap rk LM is less than 1%, the Cragg Donald Wald F (46.401) and Kleibergen–Paap rk Wald F (31.613) values are both greater than the 10% critical value (16.38) of the corresponding Stock Yogo weak instruments test, indicating that there is no invalid instrumental variable problem. The above results show that the positive effect of farmland transfer on AGTFP is not affected by endogenous problems.

### 4.4. Robustness Test

We use three methods to test robustness, including the Winsorize method, excluding the year of policy interference, and replacing the core explanatory variables; the results are shown in Table 6. First, when considering the impact of the data outliers, we use the Winsorize method to conduct the regression. That is, the formation replacing the value greater than the 95% quantile with a value of a 95% quantile, as well as a value less than the 5% quantile with a value of the 5% quantile, and displaying the result in Reg (1) in Table 6. It can be seen that the influence coefficient of the FD on AGTFP is positive, and the *p*-Value of the *t*-test is less than 1%. Second, when considering that the Chinese government issued policy documents on guiding farmland transfer in 2005, 2010, and 2014, respectively, to eliminate policy interference factors, the sample data of these three years were removed. Reg (2) in Table 6 shows that the influence coefficient of FD on AGTFP is positive, and the *p*-Value of the *t*-test is less than 1%. Third, we selected the ratio of farmland transfer to agricultural operation scale (FDA) as an alternative core variable for regression. Reg (3) in Table 6 shows that the influence coefficient of FDA on AGTFP is also positive, and the *p*-Value of the *t*-test is less than 10%, rejecting the null hypothesis that the FDA does not influence AGTFP. It indicates that the regression results in Table 4 are robust.

### 4.5. Heterogeneity Test

When considering China’s apparent natural and economic disparities, the effect of farmland transfer in different regions may exist heterogeneity. We divided the study samples into eastern, central, and western regions for heterogeneity testing [10]. The results are shown in regression (1)–regression (3) of Table 7. It can be seen that all the influence coefficients of FD on AGTFP in the eastern, central, and western regions are positive, and the *p*-Values of the *t*-test are less than 1, 1, and 5%, respectively. In addition, when considering the possible heterogeneity caused by the grain-production functional difference, we also tested the heterogeneity of those samples from major grain-producing areas and nonmajor grain-producing areas [66]. The results are shown in regression (4) and regression (5) in Table 7. It can be seen that the influence coefficients of FD on AGTFP are positive, and the *p*-Values of the *t*-test are less than 10% and 1%, respectively. The regression coefficient of farmland transfer in nonmajor grain-producing areas is bigger than that in the main grain-producing areas. This is because most of the nonmajor grain-producing areas are in the western provinces, where the population is small and the area is vast. The implementation of farmland transfer can improve the efficiency of land resource utilization by forming a land management model at an appropriate scale, thus increasing the AGTFP.

### 4.6. Mechanism Test

The mechanism test results are shown in Table 8. Regression (1) and regression (2) are the results of the interaction items of nonagricultural labor transfer (FLT) and agricultural technology utilization (ATS) with farmland transfer, respectively. It can be seen that the estimated coefficients of FD*FLT and FD*ATS are positive, and the *p*-Values of the *t*-test are less than 10% and 1%, respectively; hence, it is reasonable to reject the null hypothesis that FD does not influence AGTFP though FLT and ATS in favor of Hypothesis 2 and Hypothesis 3 that FD can increase AGTFP through FLT and ATS. It indicates that nonagricultural labor transfer and agricultural technology utilization can effectively increase AGTFP in regions where the farmland transfer is expanding. In order to further ensure the mechanism test’s robustness, this paper lists the estimated results of the intermediary effect model in regressions (3)–(7) in Table 8. Firstly, the influence coefficient of farmland transfer on AGTFP is 0.110, and the *p*-Value of the *t*-test is less than 1%; Secondly, the influence coefficients of farmland transfer on nonagricultural labor transfer and agricultural technology utilization are 0.276 and 0.026, and the *p*-Values of the *t*-test are less than 5% and 1%, respectively; finally, the core explanatory variables and intermediary variables are included in the model at the same time. Regression (5) and regression (7) are the estimated results after nonagricultural labor transfer and agricultural technology utilization are included, respectively. It can be seen that the influence coefficients of farmland transfer on AGTFP are significantly positive, and all the *p*-Values of the *t*-test are less than 1%.

The influence coefficients of nonagricultural labor transfer and agricultural technology utilization on AGTFP are 0.021 and 0.648, and the *p*-Values of the *t*-test are less than 10% and 1%, respectively. The indirect effects of nonagricultural labor transfer and agricultural technology utilization on AGTFP are about 0.006 and 0.017, respectively. It can be concluded that nonagricultural labor transfer plays an intermediary role of 5.45% in farmland transfer and affects AGTFP, while agricultural technology utilization plays an intermediary role of 15.45%. Indicating that farmland transfer can furtherly promote AGTFP by accelerating the nonagricultural transfer of the labor force and improving agricultural technology utilization.

## 5. Discussion

Farmland transfer is an important means of the rational allocation of farmland resources and is a popular research topic. Previous studies discuss the economic impact of farmland transfer. For example, Helfand and Taylor (2021) [11] believe that the expansion of agricultural operation scale brought about by farmland transfer reduces agricultural TFP (total factor productivity), while Kuang et al. (2021) [10] found that there is an “inverted u-shaped” relationship between farmland transfer and agricultural TFP. Few studies discuss the environmental impact of farmland transfer. Li et al. (2021) [16] point out that farmland transfer expands the scale of farmland management and encourage farmers to use organic fertilizer, thus improving the agricultural production environment. Wu et al. (2018) [17] and Zaehringer et al. (2018) [19] confirm this view. However, in the process of agricultural production, farmland transfer not only affects economic output but also affects the agricultural environment. Only considering the economic effect or environmental effect of farmland transfer will lead to a deviation in the research results. One of our contributions is to consider the economic effect (foreseen output) and environmental effect (undesirable output) of farmland transfer and put this into the framework of agricultural green total factor productivity (AGTFP). This approach is similar to the research of Li et al. (2021) [16] and Wu et al. (2018) [17]. However, previous research only considered agricultural nonpoint source pollution [67] or greenhouse gas emissions [49] when measuring AGTFP and used the SBM-DEA model for the measurements. We combine agricultural nonpoint source pollution and greenhouse gas emissions into the calculation of AGTFP, which can make the calculated AGTFP closer to the actual agricultural production. At the same time, we use the super efficiency SBM-DEA model to measure the AGTFP, which can solve the problem that the SBM-DEA model cannot rank the effective decision-making units, thus improving the existing research.

We further found that previous studies on farmland transfer were mainly conducted at the level of agricultural producers or agricultural families. For example, Udimal et al. (2020) [68] found that farmland transfer is conducive to improving a family’s income and a family’s welfare. Li et al. (2020) [69] found that farmland transfer can reduce the poverty vulnerability of farmers’ families; with an increase in the transfer area, the poverty alleviation effect becomes stronger. Few pieces of literature discuss the impact of farmland transfer from the macrolevel, and mainly focus on the economic growth and the utilization efficiency of farmland. For example, Shao and Zhang (2015) [37] believe that farmland transfer is conducive to alleviating the phenomenon of abandoned farmland in rural mountainous areas, thereby promoting economic growth. Fei et al. (2021) [15] found that farmland transfer is conducive to improving the utilization efficiency of farmland by using China’s provincial panel data. Our second contribution is to investigate the impact of farmland transfer on AGTFP at the macrolevel, which expands the research of Shao and Zhang (2015) [37] and Fei et al. (2021) [15]. At the same time, we further discuss the mechanism of farmland transfer affecting AGTFP from the perspective of nonagricultural labor transfer and agricultural technology utilization, providing a new perspective for the mechanism study of farmland transfer affecting AGTFP. This will help us to explore new ways to improve China’s AGTFP and green agriculture development.

This study also has some limitations: First, due to the large amounts of missing annual data for prefecture-level cities, we were unable to use the data at the city-prefecture level for the analysis. In the next study, we will try to collect panel data at the prefecture level to analyze the effect of farmland transfer. Second, although we have considered agricultural nonpoint source pollution and greenhouse gas emissions when calculating AGTFP, the types of agricultural pollution emissions may still not comprehensively cover this aspect enough. This may lead to a way of expanding the selection of pollutant indicators in future research.

## 6. Conclusions and Policy Recommendations

The existing literature lacks a macro perspective consideration of farmland transfer on agricultural green total productivity (AGTFP) in China. Based on this, we explored the impact and mechanism of farmland transfer on China’s AGTFP based on the provincial panel data from 2005 to 2020 and used a two-way fixed effect model and mediating effect model to conduct a systematic empirical test of the theoretical hypothesis. The main conclusions of this paper are as follows: (1) farmland transfer has a significant positive impact on AGTFP. This conclusion is still valid after multiple robustness tests, such as the Winsorize method, excluding years of policy interference and replacing the core explanatory variables. (2) The regional heterogeneity tests find that farmland transfer in nonmajor grain-producing areas has a more substantial role in promoting AGTFP than that in major grain-producing areas, indicating that farmland transfer has a more significant role in increasing AGTFP in relatively backward areas. (3) The mechanism test found that farmland transfer can promote regional AGTFP through nonagricultural labor transfer and agricultural technology utilization.

The above conclusions have important policy implications for improving China’s farmland transfer policy and promoting China’s green agricultural development. Firstly, empirical research shows that farmland transfer and its lag term significantly promote AGTFP. Therefore, China should further adhere to the farmland transfer policy, speed up the construction of the farmland transfer market, and attach importance to the essential role of the market in allocating agricultural production factors to improve the scale of agricultural land operation. Secondly, when considering that farmland transfer plays a more significant role in promoting AGTFP in relatively backward areas, China should flexibly adjust the farmland transfer policy, encourage and advocate the leaders of agricultural development in the back areas to vigorously carry out farmland transfer, reduce land fragmentation, and encourage the appropriate agricultural scale operation to improve AGTFP. Finally, when considering the intermediary role of nonagricultural labor transfer and agricultural technology utilization between farmland transfer and AGTFP, it is necessary to strengthen production-skills training for agricultural producers, increase investment in green agricultural technology, extend agricultural green production technology to agricultural producers, and finally promote green agricultural development.

## Figures and Tables

**Table 1 ijerph-20-02130-t001:** Specific measurement indicators of AGTFP.

Factors	Indicators	Measurement Methods	Data Sources
Input	Land	The total sown area of crops	“China Rural Statistical Yearbook”
	Labor	The total labor force in the plantation industry	“China Statistical Yearbook”
	Draft animals	Number of large livestock	“China Rural Statistical Yearbook”
	Mechanical power	Total power of agricultural machinery	“China Rural Statistical Yearbook”
	Irrigation	Actual irrigation area	“China Rural Statistical Yearbook”
	Pesticides	Pesticide usage	“China Rural Statistical Yearbook”
	Agricultural film	Amount of agricultural film used	“China Rural Statistical Yearbook”
	Fertilizer	The amount converted from fertilizer application	“China Rural Statistical Yearbook”
Output	Foreseen output	The total agricultural output value	“China Statistical Yearbook”
	Undesirable output	The agricultural comprehensive pollution index	“Calculated by the author.”
		Agricultural carbon emissions	

**Table 2 ijerph-20-02130-t002:** Descriptive Statistical of Variables.

Variable Name	Code	N	Mean	Sd	Min	Max
Agricultural green total factor productivity	AGTFP	480	1.578	0.648	0.384	4.171
The scale of farmland transfer	FD	480	1.028	1.227	0.009	6.897
Nonagricultural labor transfer	FLT	480	3.700	5.849	0.299	32.730
Agricultural technology utilization	ATS	480	0.600	0.253	0.197	1.416
Agricultural structure	INS	480	0.523	0.086	0.338	0.746
Income distribution	IND	480	2.805	0.543	1.850	4.600
Rural energy consumption	EN	480	6.539	1.171	4.007	10.618
Financial support for agriculture	FSA	480	0.238	0.388	0.012	2.038
Trade dependence on agricultural products	OPEN	480	0.301	0.361	0.016	1.696

**Table 3 ijerph-20-02130-t003:** China’s AGTFP and its decomposition from 2005 to 2020.

Year	AGTFP	AGEC	AGTC	Year	AGTFP	AGEC	AGTC
2004–2005	1.0204	0.9284	1.1112	2012–2013	1.0629	0.9995	1.0626
2005–2006	0.9926	0.9614	1.0382	2013–2014	1.0377	0.9776	1.0628
2006–2007	1.0877	1.0377	1.0613	2014–2015	0.9688	0.9566	1.0179
2007–2008	1.0711	1.0145	1.0784	2015–2016	1.0574	0.9774	1.0876
2008–2009	1.0352	1.0216	1.0475	2016–2017	0.9242	1.0068	0.9264
2009–2010	1.2066	0.9794	1.2566	2017–2018	1.1622	1.0191	1.1421
2010–2011	1.0637	0.9925	1.0785	2018–2019	1.1154	0.9797	1.1405
2011–2012	1.0985	1.0239	1.1003	2019–2020	0.8877	1.0217	0.8696
Eastern	1.0460	0.9954	1.0535	Central	1.0435	1.0010	1.0579
Western	1.0574	0.9864	1.0888	Average	1.0495	0.9936	1.0676

Note: The average value in the table is the arithmetic mean.

**Table 4 ijerph-20-02130-t004:** Estimation results of FD on AGTFP.

Variable	Coefficient	t-Value
FD	0.110 ***	4.21
INS	5.186 ***	8.54
IND	−0.358 ***	−4.04
EN	0.230 ***	3.58
FSA	0.138 ***	2.85
OPEN	−0.364 **	−2.36
Cons_	−1.666 **	−2.33
Time effect	Yes	
Regional effect	Yes	
N	480	
R^2^	0.569	

Note: **, *** in the table indicate significance at 5%, and 1%, respectively.

**Table 5 ijerph-20-02130-t005:** Regression results of endogenous regressions.

Variable	(1) Reg 1	(2) Reg 2	Reg 3	Reg 4
L.FD	0.065 **			
	(2.42)			
FD		0.036 **		0.265 ***
		(1.97)		(3.23)
IV			0.430 ***	
			(6.81)	
Cons_	−1.494 *	−0.072		
	(−1.82)	(−0.42)		
Kleibergen–Paap rk LM			25.760 ***	
Cragg-Donald Wald F value			46.401	
Kleibergen–Paap rk Wald F value			31.613	
Control variable	Yes	Yes	Yes	Yes
Time effect	Yes	Yes	Yes	Yes
Regional effect	Yes	Yes	Yes	Yes
N	450	450	480	480
R^2^	0.541		0.435	0.359

Note: *, **, and *** in the table indicate significance at 10, 5, and 1%, respectively. The value in the parentheses of Reg (1), Reg (3), and Reg (4) is the *t*-value; the value in the parentheses of Reg (2) is the *z*-value. L.FD indicates that the variable FD lags 1 period.

**Table 6 ijerph-20-02130-t006:** Robustness test results.

Variable	Reg (1)	Reg (2)	Reg (3)
	Winsorize Treatment	Eliminate Policy Interference	Replace Explanatory Variables
FD	0.108 ***	0.095 ***	
	(4.72)	(3.30)	
FDA			0.026 *
			(1.92)
Cons_	−1.556 **	−1.465 *	−2.210 ***
	(−2.32)	(−1.67)	(−3.10)
Control variable	Yes	Yes	Yes
Time effect	Yes	Yes	Yes
Regional effect	Yes	Yes	Yes
N	480	390	480
R^2^	0.587	0.573	0.555

Note: *, **, and *** in the table indicate significance at 1, 5%, and 1%, respectively. The value in parentheses is the *t*-value.

**Table 7 ijerph-20-02130-t007:** Heterogeneity test results.

Variable	Economic Functional Area			Food Production Functional Area	
	(1) East	(2) Central	(3) Western	(4) Main-Grain Producing	(5) Non-Major Grain Producing
FD	0.142 ***	0.164 ***	0.144 **	0.070 *	0.182 ***
	(3.21)	(4.07)	(2.04)	(1.83)	(2.69)
Cons_	3.128 ***	−6.107 ***	0.587	0.818	0.107
	(4.38)	(−4.27)	(0.61)	(1.10)	(0.11)
Control variable	Yes	Yes	Yes	Yes	Yes
Time effect	Yes	Yes	Yes	Yes	Yes
Regional effect	Yes	Yes	Yes	Yes	Yes
N	176	128	176	208	272
R^2^	0.504	0.682	0.601	0.585	0.572

Note: *, **, and *** in the table indicate significance at 10, 5, and 1%, respectively. The value in parentheses is the *t*-value.

**Table 8 ijerph-20-02130-t008:** Mechanism test results.

Variable	Reg (1)	Reg (2)	Reg (3)	Reg (4)	Reg (5)	Reg (6)	Reg (7)
	AGTFP	AGTFP	AGTFP	FLT	AGTFP	ATS	AGTFP
FD*FLT	0.021 *						
	(1.69)						
FD*ATS		0.282 ***					
		(3.12)					
FD	0.106 ***	−0.028	0.110 ***	0.276 **	0.156 ***	0.026 ***	0.135 ***
	(2.87)	(−0.45)	(4.21)	(2.13)	(6.16)	(3.68)	(5.35)
FLT					0.021*		
					(1.76)		
ATS							0.648 ***
							(3.87)
Cons_	0.380	0.270	−1.666 **	10.184 ***	0.266	0.685 ***	0.001
	(0.82)	(0.58)	(−2.33)	(4.23)	(0.56)	(5.30)	(0.00)
Control variable	Yes	Yes	Yes	Yes	Yes	Yes	Yes
Time effect	Yes	Yes	Yes	Yes	Yes	Yes	Yes
Regional effect	Yes	Yes	Yes	Yes	Yes	Yes	Yes
N	480	480	480	480	480	480	480
R^2^	0.560	0.567	0.569	0.110	0.560	0.213	0.572

Note: *, **, and *** in the table indicate significance at 10, 5, and 1%, respectively. The value in parentheses is the *t*-value.

## Data Availability

All the data are obtained from the China Statistical Yearbook, China Rural Statistical Yearbook, Statistical Data of National Rural Economy, Annual Statistical Report of China’s Rural Operation and Management, Annual Statistical Report of China’s Rural Policy and Reform, Statistical Yearbook of China Population and Employment and China Agricultural Products Trade Development Report (2006–2021). It is available on request from the corresponding author.

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
