# Peer review of "The Effect of Farmland Transfer on Agricultural Green Total Factor Productivity: Evidence from Rural China"

_ijerph, 2023, doi:10.3390/ijerph20032130_

Round 1
Reviewer 1 Report
The authors address an interesting research problem in their manuscript.
I have some substantial comments and some formal remarks that the authors can find in the attached pdf file.

Author Response
Dear Reviewer,
Happy New Year to you and your family. Thank you so much for allowing us to revise and resubmit our paper. My co-authors Guoqun Ma, Xiaopeng Dai and I are pleased to submit a revised version of the paper titled "The effect of farmland transfer on agricultural green total factor productivity: Evidence from rural China" for consideration for possible publication in the section of Article of International Journal of Environmental Research and Public Health. This submission includes a manuscript, the response and the highlights. We have made every possible effort to use the feedback to improve the manuscript. We believe that the guidance has resulted in a substantially improved paper. We would like to thank you again for your support and encouragement.
We look forward to your feedback on this revised submission.
Sincerely,
Guoqun Ma, Xiaopeng Dai and Yuxi Luo

Reviewer 2 Report
This paper analysis the relationship between farmland transfer and agricultural green total factor productivity, which provide an interesting study. Some suggestions are as follows.
1. This paper lacks the literature part relate to relationship between farmland transfer and agricultural green total factor productivity
2.This paper uses the Super-efficiency DEA-SBM model and the GML index to calculate AGTFP. Why use two methods to caculate AGTFP? This paper use d which method, Why?
3. How did the author get the agricultural comprehensive pollution index, the author should give out relate information
4.Page 11, Line 340, please carefully check the paper
Author Response
Dear Reviewer,
Happy New Year to you and your family. Thank you so much for allowing us to revise and resubmit our paper. My co-authors Guoqun Ma, Xiaopeng Dai and I are pleased to submit a revised version of the paper titled "The effect of farmland transfer on agricultural green total factor productivity: Evidence from rural China" for consideration for possible publication in the section of Article of International Journal of Environmental Research and Public Health. This submission includes a manuscript, the response and the highlights. We have made every possible effort to use the feedback to improve the manuscript. We believe that the guidance has resulted in a substantially improved paper. We would like to thank you once again for your support and encouragement.
We look forward to your feedback on this revised submission.
Sincerely,
Guoqun Ma, Xiaopeng Dai, and Yuxi Luo

Round 2
Reviewer 1 Report
I can consider myself satisfied limitedly with the reply by authors to my specific remarks in their cover letter and the way they have revised the manuscript accordingly in the points I have signalled.
Nevertheless, the section about results (4) – which I have deemed almost unreadable in my previous report, still raises some serious concerns, that I address in my critical remarks in the following – along with suggestions for minor edits. Authors should revise the way the whole statistical results are presented; there are inconsistencies in the data included in tables and the discussion made in the text; some paragraphs are still too much involved so their meaning remains obscure.
The use of jargon and the prevalence of the structure of software output in driving the explanations do not help to clarify what path of analysis led the authors to the conclusions. The further developments mentioned at the end of the Discussion section shade doubts on the linear analysis (even if in the previous section results are deemed ‘robust’) so reducing the reader’s trust in the reported findings. I still wonder if the inclusion of the word suggestions in the title of section 6 is appropriate since it seems that the authors are offering some indication for policy-making instead of suggestions for further research.
Please, find in the attached PDF file my detailed remarks.

Author Response
Dear Reviewer,
We are grateful to your valuable comments and suggestions. Based on your review report, we did a point-by-point rechecks and revisions. A detailed response is attached. We hope that our revised manuscript and response can make you satisfied.
Again, thanks for your time and comments. We look forward to hearing from you.
Sincerely,
Guoqun Ma, Xiaopeng Dai, and Yuxi Luo

Reviewer 2 Report
The author solved all the problems
Author Response
Dear Reviewer,
Thank you for allowing this paper to be published in the International Journal of Environmental Research and Public Health. We further improved the manuscript, making our empirical results easier for readers to understand.
Again, thanks for your time and comments.
Sincerely,
Guoqun Ma, Xiaopeng Dai, and Yuxi Luo